# Singlet Oxygen *In Vivo*: It Is All about Intensity—Part 2

**DOI:** 10.3390/jpm13050781

**Published:** 2023-04-30

**Authors:** Steffen Hackbarth, Shanghui Gao, Vladimír Šubr, Lisheng Lin, Jakob Pohl, Tomáš Etrych, Jun Fang

**Affiliations:** 1Institute of Physics, Photobiophysics, Humboldt University of Berlin, Newtonstr. 15, 12489 Berlin, Germany; lslin@fjnu.edu.cn (L.L.); jakob.pohl@physik.hu-berlin.de (J.P.); 2Laboratory of Microbiology and Oncology, Faculty of Pharmaceutical Sciences, Sojo University, Kumamoto 860-0082, Japan; gaoshanghui94@gmail.com (S.G.);; 3Institute of Macromolecular Chemistry, Czech Academy of Sciences, Heyrovského nám. 2, 16200 Prague, Czech Republic; subr@imc.cas.cz (V.Š.);; 4Fujian Provincial Key Laboratory of Photonics Technology, Fujian Normal University, Fuzhou 350007, China

**Keywords:** photodynamic therapy, singlet oxygen, time-resolved phosphorescence, illumination intensity

## Abstract

Recently, we reported induced anoxia as a limiting factor for photodynamic tumor therapy (PDT). This effect occurs *in vivo* if the amount of generated singlet oxygen that undergoes chemical reactions with cellular components exceeds the local oxygen supply. The amount of generated singlet oxygen depends mainly on photosensitizer (PS) accumulation, efficiency, and illumination intensity. With illumination intensities above a certain threshold, singlet oxygen is limited to the blood vessel and the nearest vicinity; lower intensities allow singlet oxygen generation also in tissue which is a few cell layers away from the vessels. While all experiments so far were limited to light intensities above this threshold, we report experimental results for intensities at both sides of the threshold for the first time, giving proof for the described model. Using time-resolved optical detection in NIR, we demonstrate characteristic, illumination intensity-dependent changes in signal kinetics of singlet oxygen and photosensitizer phosphorescence *in vivo*. The described analysis allows for better optimization and coordination of PDT drugs and treatment, as well as new diagnostic methods based on gated PS phosphorescence, for which we report a first *in vivo* feasibility test.

## 1. Introduction

Singlet oxygen (^1^O_2_) is the first electronically excited state of molecular oxygen. It plays a major role in many biological processes and is well known as the intended main mediator of photodynamic therapy (PDT) [1,2]. In 1938, Kautsky proposed the energy transfer from a photosensitizer (PS) in its triplet state to molecular oxygen as the generation pathway [3], and it was not until 1968 that Snelling [4] was able to prove it. Today, there is general agreement that local generation of ^1^O_2_ might be one possible way to treat cancer and other diseases [1,2,5,6,7]. All it takes is an appropriate PS, illumination at a useful wavelength, and some oxygen [8]. Oelckers et al. [9] first reported direct observation of ^1^O_2_ in biological material. During measurements in erythrocyte ghosts, they could attribute a part of the detected signal to ^1^O_2_ in the membrane. However, at that time, NIR detectors were analogue devices and the Ge PIN Diode of Northcoast the gold standard. Analogue detection has some disadvantages at low intensities [10,11], which made it seem impossible to observe ^1^O_2_
*in vivo* [12].

In recent years, detection has become much easier with commercially available NIR-PMTs such as Hamamatsu’s H10330-45. Several authors reported ^1^O_2_ observation *in vitro* or *in vivo*, among others Jarvi et al. [13], Pfitzner et al. [14], Schlothauer et al. [15], Li et al. [16], and Scholz et al. [17]. A first indication of how useful ^1^O_2_ monitoring might be for clinical application was given by Wilson et al. [18] who observed a correlation between the detected steady ^1^O_2_ phosphorescence intensity and the treatment response of implanted sarcoma in mice in a window chamber model. Until now, time-resolved measurements have been limited to skin or blood vessels, even if the authors were sometimes unaware of this.

For the background to the work presented here, we strongly recommend reading two of our recent papers on this topic [19,20]. In short, the ^1^O_2_-mediated photodynamic effect is based on chemical reactions between the excited molecular oxygen and biological quenchers, mainly proteins. It is precisely this reaction which consumes both oxygen and quenchers. We already demonstrated quencher consumption more than a decade ago [21,22]. While the proof was given for an *in vitro* system at that time, quencher consumption will follow the same dynamics *in vivo*. The highly efficient quenching in cells drastically shortens the diffusion length of ^1^O_2_ well below cellular dimensions, making the places of generation and quenching part of the same cell [23]. This happens both *in vitro* or *in vivo*, because unlike oxygen, proteins cannot be easily replaced by diffusion from the outside.

For oxygen, the situation is completely different. In well-done *in vitro* experiments, sufficient oxygen supply is guaranteed via diffusion from the growth medium with which every cell has direct contact. *In vivo*, oxygen supply is mainly provided by blood vessels. Oxygen diffuses out of the vessel following the concentration gradient, slowed down only by the reduced oxygen solubility and diffusion in the vessel wall [24]. Consequently, the oxygen supply decreases with distance from the blood vessels, as impressively shown in [25] by the nearly linear dependency between vessel diameter and the maximum distance from the vessel at which cells survived. The normal metabolism of tumor cells is much higher than that of normal cells [26], resulting—even without additional oxygen consumption—in a lower pO_2_ in tumors and all its consequences, such as VEGF overexpression and higher vascularization with less perfect vessel walls, one of the causes of the EPR effect [27]. PDT treatment, even at moderate illumination, causes strong additional oxygen consumption. If the resulting overall oxygen consumption exceeds the supply, the local oxygen concentration drops to nearly zero within a short distance from the blood vessel.

During *in vivo* experiments in a mouse model with systemically applied PS, we found stronger ^1^O_2_ phosphorescence signals originating in normal tissue with low PS concentration than in tumors with significantly higher PS concentration and much stronger fluorescence signals [16]. Following the Krogh geometry and the vascularization parameters described in [22,28], this effect can be theoretically illustrated in a simplified model of the tumor tissue. Based on these theoretical considerations, we estimated the radial pO_2_ profiles around a vessel for given illumination intensities [16] which were then used to calculate estimated signal kinetics [17]. We put in experimental parameters such as the intensity ratio of ^1^O_2_ and PS phosphorescence at 1270 nm, the triplet decay time of pyropheophorbide-a (pP) in oxygen-free micellar suspensions (around 415 µs), the ^1^O_2_ kinetics in blood (as determined in a HET-CAM [29] model) and drug concentration both in the tumor and in plasma as well as published parameters for oxygen solubility and diffusion in different types of tissue [21,30]. The theoretical model predicted the observed experimental results at the chosen illumination intensity, attributing the kinetics to the emerging anoxia. When we reduced the illumination intensity in these theoretical considerations by two orders of magnitude, anoxia could no longer be observed and the majority of ^1^O_2_ was generated in the surrounding tissue.

In practice, reducing the intensity comes at the cost of prolonging the illumination time by the same factor to achieve the same dose. At normal operation of our setup, a measurement will take 30 s at a pulse rate of 12 kHz. To minimize the time, mice had to be kept narcotized, in most experiments the intensity was decreased by a factor of 32 (16 min at 375 Hz), which turned out to be low enough to allow for the experimental verification of our proposal on PDT-induced anoxia in tumor tissue. Only in one experiment, we varied the illumination intensity by a factor of 128 (64 min at 94 Hz).

While we have recently presented experimental results for *in vivo* measurements at high intensities [17], which match the predicted kinetics, the experimental realization of measurements at low intensities was not possible at that time. In this work, we exploit the advantage of time-resolved detection that noise signals can be discriminated according to their time tag. We reach ^1^O_2_ phosphorescence signals at average illumination intensities as low as 0.5 mW/cm² at unprecedented signal to noise ratio.

## 2. Materials and Methods

*N*-(2-hydroxypropyl)methacrylamide polymers loaded with pyropheophorbide-a (HPMA-pP) methacryloyl chloride, 1-amino-propan-2-ol, 2,2′-azo*bis*isobutyronitrile (AIBN), 6-amino hexanoic acid (Ahx), *tert*-butoxycarbonyl hydrazide (NH_2_NH-Boc), *N*-ethyl-*N*′-(3-dimethylaminopropyl)carbodiimide hydrochloride (EDC.HCl), carbon disulfide, ethanethiol, sodium hydride (60% dispersion in mineral oil), 4,6-trinitrobenzene-1-sulfonic acid (TNBSA), pentafluorophenol, 4-(dimethylamino)pyridine (DMAP), *tert*-butanol (*tert*-BuOH), *N,N*-dimethylacetamide (DMAA), dichloromethane (DCM), and dimethyl sulfoxide (DMSO) were obtained from Merck (Czech Republic). Pyropheophorbide-a was purchased from Frontier Scientific^®^ and 2,2′-azo*bis*(4-methoxy-2,4-dimethylvaleronitrile) (V-70) from Wako Chemicals (Germany). All of the other chemicals and solvents were of analytical grade. 

Monomers *N*-(2-hydroxypropyl)methacrylamide (HPMA) and *N*-(*tert*-butoxycarbonyl)-*N*′-(6-methacrylamidohexanoyl)hydrazine (Ma-Ahx-NHNH-Boc) were synthesized according to the literature [31,32] (Figure 1).

The chain transfer agent S-2-cyano-2-propyl S′-ethyl trithiocarbonate (CTA-AIBN) and 4-cyano-4-(ethylthiocarbonothioylthio)pentanoic acid (CTA-COOH) were synthesized as previously described [33,34].

*Tert-butyl N-[2-[(4-cyano4-ethylsulfanylcarbothioylsulfanyl-pentanoyl)amino]ethyl] carbamate (CTA-ED-Boc)* was prepared by the reaction of CTA-COOH with *N*-Boc-1,2-diaminoethane (ED-Boc) in the presence of *N*-(3-dimethylaminopropyl)-*N*′-ethylcarbodiimide hydrochloride (EDC.HCl) in dichloromethane (DCM). 

CTA-COOH (1.0 g; 3.83 mmol) and ED-Boc (0.68 g; 4.22 mmol) were dissolved in 20 mL of DCM and EDC.HCl (1.05 g; 5.48 mmol) was added. Reaction mixture was stirred for 3 h at room temperature. Reaction mixture was extracted with distilled water (2 × 10 mL) and with 0.2 M NaHCO_3_ (2 × 10 mL) and dried with anhydrous Na_2_SO_4_. DCM was evaporated and oily residue was dissolved in ethyl acetate 20 mL and pure CTA-ED-Boc was obtained by using column chromatography on silica gel in ethyl acetate. Ethyl acetate was evaporated in vacuo and pure CTA-ED-Boc was obtained as orange oily product. Yield was 1.19 g (76.3%). HPLC analysis on Chromolith RP18e column with PDA detection at 305 nm showed single peak with retention time 10.4 min. 

ESIMS: *m/z* (M) + calculated for C_16_H_27_N_3_O_3_S_3_ 405.60 found m/z [M + Na] + 427.74.

The polymer precursor poly(HPMA-*co*-Ma-Ahx-NHNH_2_) was prepared by the controlled radical reversible addition–fragmentation chain transfer (RAFT) copolymerization of HPMA and Ma-Ahx-NHNH-Boc in *ter*t-BuOH using CTA-AIBN and the initiator 2,2′-azo*bis*(4-methoxy-2,4-dimethylvaleronitrile) (V-70) at 30 °C for 72 h. The molar ratio of monomer/CTA/V-70 was 360/2/1, and the molar ratio of monomers HPMA/Ma-Ahx-NHNH-Boc was 90/10 [35].

The semitelechelic polymer precursor poly(HPMA)-NH-Boc was prepared by using RAFT polymerization of HPMA in *ter*t-BuOH in the presence of CTA-ED-Boc and 2,2′-azo*bis*(4-methoxy-2,4-dimethylvaleronitrile) at 30 °C for 72 h [17]. Trithiocarbonate end groups were removed via reaction with an excess of AIBN, and Boc groups were thermally removed in Q-H_2_O [36,37].

The copolymer s-HPMA-pP was synthesized by reaction of semitelechelic poly(HPMA)-NH_2_ with the pentafluorophenyl ester of pP [16]. This single-loaded polymer is non-cleavable but with a very high activity in water and in tissue. In aqueous solutions containing detergents (0.5% Triton or Tween), this conjugate has a ^1^O_2_ quantum yield comparable to monomeric pP, hence 0.52 +/− 0.05. In water, we determined still 0.20 +/− 0.05. This is the highest value of all HPMA-pP conjugates that we reported so far. As we could show before, it accumulates well in the tumor and remains for many days [17]. On the other hand, non-cleavable HPMA copolymers circulate for many days in the blood plasma and accumulate in the liver after systemic application to mice [38]. This is likely a disadvantage for medical applications, as high contrast and selectivity is aspired in any case. However, at this stage, we appreciate the long residence time in the tumor and the high singlet oxygen quantum yield even in tissue, which is advantageous for diagnostic purposes as we will show later.

The copolymer h-HPMA-pP includes a pH-sensitive hydrolytically degradable hydrazone bond and was described in detail in [39]. Cleavable conjugates potentially have the major advantage of faster clearing, from the tumor [29] as well as from liver. While it also has a ^1^O_2_ quantum yield comparable to the free dye in aqueous solutions containing 0.5% detergents, in water, there is only very little singlet oxygen generation (around 0.05) observable. This is due to the fact that this polymer is loaded statistically with 3–4 PS moieties each. The quite flexible and long linkers allow stacking of the attached pPheo, resulting in excitonic interaction and non-radiative deactivation in water [40].

Both conjugates exhibit the so-called EPR effect (enhanced permeability and retention with hydrodynamic diameters of around 20 nm (s-HPMA-pP) and around 16 nm (h-HPMA-pP) in water. It is not surprising that the multi-loaded polymer, even though it has a higher molecular weight, 26,400 Da, compared to 11,500 Da, has a smaller hydrodynamic diameter, as the stacking of the attached pPheo moieties likely folds the polymer backbone to a more compact size.

The mouse sarcoma S180 solid tumor model was used in this study, just like in our previous studies [16,17]. In brief, 6-week-old (30–35 g) male ddY mice were purchased from SLC, Shizuoka, Japan. They stayed, with free access to water and food, at 20–22 °C and 45–50% humidity with a 12 h light/dark cycle. Sarcoma S180 cells grown as ascetic form in the peritoneal cavity of ddY mice were implanted subcutaneously in the dorsal skin of ddY mice (2 × 10^6^ cells). After the tumor grew to 8–10 mm in size, the polymer conjugate was injected into the tail vein. We used 5 mg/kg, according to the attached pP, diluted in physiological saline solution. Other than slight shaving of the area of interest and anesthesia with isoflurane, the animals were not treated in any further way.

Time-resolved phosphorescence of pP and the generated ^1^O_2_ were measured with a TCMPC NIR detection system (SHB Analytics, Berlin, Germany) with fiber adapter for a furcated quartz fiber (Ceram Optec, Bonn, Germany) used for simultaneous excitation and detection. For detection around 1270 nm with a HWFM of 40 nm, we used a modified version of the NIR-PMT H10330-45 (SHB Analytics/Hamamatsu) with increased etendue for more than three-fold better detection in voluminous samples compared to the original detector. For detection in the spectral range of 850–950 nm, we chose a PMT Module H14121-61 (Hamamatsu, Hamamatsu, Japan). A custom-built laser, based on the laser diode Red 65X at around 658 nm (Necsel, Cypress, Milpitas, CA, USA), was driven in pulsed operation with ~350 ns/0.58 µJ per shot resulting in an average intensity of 7 mW when driven at 12 kHz. Every measurement consists of 360,000 pulses and hence 210 mJ.

The fiber tip was positioned using a custom-made 2D scanning unit with 0.3 mm spatial resolution either with the tip in direct contact to the skin or, for scanning experiments, in a confocal manner focusing the excitation at the skin and collecting the emission across the same lens. In this work, the vertical position was not yet corrected for the silhouette of the mouse. However, distance variation mainly influences the intensity but not the determined kinetics, at least not at the highest intensity (12 kHz). In each case, excited and observed tissue volume automatically coincide.

The average illumination intensity was adjusted by controlling the pulse repetition frequency. A custom-built frequency generator in combination with a 7-bit counter as switchable frequency divider together with a pulse shaper delivered a stable pulse train of constant pulse duration and adjustable frequency, ranging from 12 kHz down to 94 Hz. The electronically driven laser diode works at a duty cycle of just 0.4% or less. This way, no thermal or saturation effects occur and the pulses are identical in all settings. Regarding the oxygen consumption, we can consider this pulsed excitation as quasi cw as the pulses are short enough to guarantee one excitation per molecule in maximum, and they are low enough (≤60 W/cm² during the pulse) to avoid ground state depletion effects. The detection time window was 80 µs, independent from the repetition rate. Therefore, the signal to noise ratio is independent of the repetition rate.

## 3. Results and Discussion

When we are observing photosensitization *in vivo* in the wavelength range around 1270 nm, we mainly see two signal contributions of the system under investigation: ^1^O_2_ and PS phosphorescence [17]. For the latter one, naturally only the long wavelength tail of the emission is detected. Apart from those signal contributions, there is always the typical short-term artifact (STA) and thermal background noise. The STA comprises non-discriminated signal components from fluorescence and related emissions as well as black body emission from local hot spots after vibrational relaxation (e.g., in case of excitonic interactions between highly concentrated dye molecules in tumors). However, the STA usually decays within a few µs. For the remainder of the detection time window (80 µs), only low thermal noise, comparable to that at room temperature (less than 30,000 dark counts during 360,000 cycles for all 4096 channels), could be detected during our *in vivo* measurements. This background is permanently controlled during our experiments. Whenever the pulse frequency is low enough (below 500 Hz), the time separation of two consecutive excitation pulses is five times the PS phosphorescence in absence of oxygen. Then, the first channels report the dark counts, since for technical reasons, the laser pulse is delayed relative to the first detection channel by 0.5 µs.

Whenever illumination is strong enough—and thus photosensitization in the tumor is exceeding the oxygen supply—the ^1^O_2_ generation is limited to the blood and vessel wall, with the pO_2_ and likewise ^1^O_2_ generation dropping sharply outside the vessel wall [16,17]. Of course, there will be differences in the photophysical behavior of PSs dependent on their exact localization and proximity of other PSs for energy transfer processes. However, we may assume that the average phosphorescence quantum yield of tissue located PS under anoxia is independent from the selected cell. Therefore, this phosphorescence is a measure for the concentration of the tissue-located dye. For a more reliable and robust quantification, we proposed a simplified analysis by summing up all signal counts within a time frame of 40–80 µs after excitation (Figure 2). Several measurements during the first 72 h after systemic application of the two investigated conjugates s-HPMA-pP and h-HPMA-pP reveal a completely different time development of their tumor accumulation. For s-HPMA-pP, accumulation increases strongly over the first 24 h and further increases up to 27 h at a lower rate, exhibiting the typical behavior expected for a drug relying on the EPR effect. In contrast, the tumor concentration of h-HPMA-pP already starts to decrease after a few hours. The only possible explanation is that the intended cleaving of the pP from the copolymer is already taking place during the first hours after administration, and pP is partly washed out of the tumor due to its low molecular weight. There is little variation in pP concentration for different mice carrying similar tumors after identical drug administration. Technically, there are the two main results: (1) obviously time-resolved detection at 1270 nm is a good measure to follow the local drug concentration in tumors and (2) the drug uptake of our tumor model is nearly identical for a sarcoma of the same size. 

The latter one is of importance for the next result, showing signal kinetics at 1270 nm from the very first measurement at each of the tumors, 6 h after systemic application of 5 mg/kg of h-HPMA-pP, at pulse frequencies ranging from 12 kHz down to 94 Hz (Figure 3).

There are big differences in the signal kinetics within the first 10 µs after excitation, but as explained in [16,17], that can be attributed to ^1^O_2_ phosphorescence from blood and vessel wall. With one exception, this signal component is bigger for lower excitation intensities. However, depending on the microstructure of the investigated tumors, the signal contribution of bigger blood vessels in the tumor or vessels in surrounding tissue that are illuminated by the strongly scattered light may vary.

If we want to use this signal as a measure for drug concentration in the blood plasma in the future, we need to find a way to ensure reproducible positioning of the detection fiber at the investigated vessel.

As mentioned before, the detection of the PS phosphorescence from the tumor tissue is quite robust. Signal variations for repeated measurements placing the fiber at the center of the same tumor are below 10%. Another 10% uncertainty should be added for the drug accumulation, resulting in a maximum quantitative uncertainty of 20%. Assuming these uncertainties, the results in Figure 3 show that a lower excitation intensity reduces the PS phosphorescence. 

Once pP is taken up by a cell or cleaved from the polymer backbone, it is very likely membrane-located and, therefore, has comparable fluorescence and ^1^O_2_ efficiency independent of the copolymer used for delivery and independent of the place and type of the cell. In addition, the vast majority of the signal contribution in the 40–80 µs time window originates from the tissue outside the vessel wall. In this tissue, the diffusion of oxygen is much better compared to vessel walls, so the oxygenation of the tissue outside the vessel wall is decreasing with distance from the vessel, but within a small range [16]. In most of this volume, we therefore may extend the just mentioned statement on similar values for phosphorescence intensity of PS to all types of luminescence and the triplet decay time. The shorter this decay time, the higher the percentage of ^1^O_2_ phosphorescence. As determined before, the detection efficiency of our equipment for ^1^O_2_ phosphorescence is lower compared to pP phosphorescence [16]. When looking at these previously published data, keep in mind that the signal amplitude of an exponential decay is normalized by the decay time. Therefore, shortening of the triplet decay time (within the relevant range) results in lower counts in the time window 40–80 µs. The average counts in this time window can therefore act as measure of the extent of hypoxia/anoxia in the observed tissue. Comparing this average count for different excitation intensities (Figure 3, insert) allows us to conclude that lower excitation intensity results in the reduction in PDT-induced anoxia.

As already established (Figure 2), 72h after injection, the tumor accumulation of the two drugs is quite different. However, in each case the signal kinetics changed to shorter decay times at lower excitation intensity. This effect is much stronger for s-HPMA-pP, which, due to higher concentration, potentially consumes more oxygen than h-HPMA-pP at identical illumination intensity (Figure 4).

To evaluate the effect of decreased illumination intensity, the tails of phosphorescence signals around 1270 nm of s-HPMA-pP for illumination at 12 kHz and 375 Hz were fitted double-exponentially, resulting in decay times of 6,5 +/− 1 µs and 240 +/− 50 µs at 12 kHz and 7.7 +/− 0.5 µs and 78 +/− 10 µs at 375 Hz. We assign the longer decay time to the average phosphorescence lifetime of pP in the tissue in the absence of oxygen or at low pO_2_. Even at 12 kHz, this value in tissue is shorter than in oxygen-free micellar solution (415 µs). However, the detected signal, unlike in the stirred *in vitro* sample, is a mixture of signals from origins with varying conditions, which a double exponential fit just cannot describe sufficiently. Nevertheless, the change in kinetics induced by the reduction in the pulse frequency is striking.

The following rate equation can be used to estimate the local pO_2_ in the tissue based on the determined decay time [41]:(1)1τT=1τ0+k∆·pO2,
where *τT* is the measured triplet decay time, *τ_0_* is the triplet decay time in the absence of oxygen, and *k_Δ_* is the first order rate constant in dependence of *pO_2_*, describing the PS triplet quenching by oxygen resulting mainly in ^1^O_2_. Because of the aforementioned results, we approximate *τ_0_* as 240 µs and calculate *k_Δ_* based on a measured phosphorescence decay time of 3.5 +/− 0.2 µs, determined in cells *in vitro* at 37 °C, stirred under 10% oxygen [42].

A triplet decay time of 78 µs as determined for 375 Hz pulse rate corresponds to around 0.003 atm or 300 Pa. This is below the value reported to be typical for tumor tissue without PDT treatment [23], but clearly above zero. This proves that the detected signal comprises ^1^O_2_ phosphorescence from the tumor tissue further away from the vessel. 

We also measured PS phosphorescence at its maximum in the region of 930 nm as reported in [43]. The PS phosphorescence signal at 12 kHz excitation rate has a temporal shape that is very comparable to the corresponding ^1^O_2_ signal (Figure 5). The double-exponential tail fit yields 6.7 +/− 1.0 µs and 150 +/− 50 µs. At an excitation rate of 375 Hz, the decay time (80 +/− 10 µs) is identical to the one determined at 1270 nm.

The value of the shorter triplet decay time in each case corresponds well to the ^1^O_2_ signal decay time that we know from PS in blood. We are, however, not sure if the signal fraction with faster decay purely originates from PS in blood or if parts of this signal component are STA. All faster signal components are nearly identical (including amplitudes) at both excitation rates. Even if we subtract the fits from the raw data, the remaining short-term signals are identical (Figure 5, insert). 

The important message, however, is that the slow component detected in the range 850–950 nm is obviously PS phosphorescence originating from the tumor tissue. The huge error margin for the long decay times in all cases results from the detection time-window of just 80 µs, which we will increase for future measurements. 

After experimentally confirming our hypothesis on PDT-induced anoxia and the oxygen supply threshold in principle, we also want to compare the determined signal kinetics at 1270 nm with kinetics as predicted by our theoretical simulation [17] quantitatively.

Figure 6 (left) shows the comparison of theory and experiment for 2 mW/cm² or 375 Hz. The calculated radial pO_2_ distribution around a vessel as described in [16] allows the calculation of the expected ^1^O_2_ and PS phosphorescence kinetics at each position and summation of an expected signal as shown in [17]. For the PS in blood, we set the kinetics parameters equal to those determined before [26]. As the measurement was performed 72 h after drug injection, we assume an even PS distribution in the tissue and wall. The results of this simulation were now scaled to fit the experimental data. We factored in the relative concentration between active PS in blood to that in tissue and found it to be around 5.0 for 375 Hz. This means that even after 72 h, a large amount of the drug is circulating in the blood, and the majority of the detected ^1^O_2_ phosphorescence originates from the blood, even at low pulse rates. Nevertheless, a small part of the slower signal component can be assigned to ^1^O_2_ phosphorescence coming from the tissue and vessel wall (Figure 6).

We did both measurements (at 12 kHz and at 375 Hz) exactly at the same spot and directly one after the other. Therefore, we expected to get the same scaling parameter for the signal coming from blood for 12 kHz pulse frequency (Figure 6 right). To our surprise, this was not the case. The theoretical ^1^O_2_ signal from the blood at 12 kHz requires scaling with 1.1, just as if there would be around 80% less drug in the blood compared to the 375 Hz detection. It seems there is much less oxygen available in the tumor vessels than we expected, and strong illumination results in oxygen depletion even in the blood.

While for 375 Hz, experiment and theory match without any modification, in Figure 6, the experimental data for 12 kHz are compared with the simulation for 6 kHz. These data give a slightly better match, even though being very similar to the data for 12 kHz. This may have different reasons, like stronger light scattering than estimated, which reduces the local excitation intensity or simply the fact that we still use a very simple Krogh model to describe the investigated tumors. 

One should be aware of what information the model used can and cannot give. The geometry is very simple. We assume a uniform drug distribution over the whole volume. The observed time window is shorter than the decay times, and the noise is still prominent. Therefore, the model cannot provide more than a first validation. 

However, since the fit is based on measured decay times and the relative contributions of PS and ^1^O_2_ phosphorescence are related to these times, validation in principle is possible. In the tissue from which the vast majority of the slow decaying PS phosphorescence originates, oxygen can diffuse well, so variations in drug concentration do barely affect the kinetics of this signal contribution.

Most importantly, the model can distinguish both qualitatively and quantitatively between the slow kinetics in anoxic regions and the fast kinetics in the blood. It is precisely this that allows the lower ^1^O_2_ signal from the blood to be identified under strong illumination, which then proves anoxia even in the blood vessels. Whether the drug in the blood is protein-bound or not, or what quantum yield it has due to the conditions, is irrelevant for our analysis, as we are simply comparing the signals from exactly the same spot at different intensities.

Despite oxygen depletion in the vessels at 12 kHz, for most of the short ^1^O_2_ signal components in blood, we determined decay times below 10 µs. A possible explanation is that blood oxygenation between 80% and 20% stabilizes the pO_2_ in the aqueous part of the blood between 6 and 2 kPa. At blood oxygenation levels below 20%, the pO_2_ drops much faster and does not contribute that much to the signal. For pheophorbide a in liposomes in water at room temperature with a pO_2_ of 20 kPa, we once determined a triplet decay time of 1.85 µs [44]. The diffusion coefficient of oxygen in water at 37° is around 25% higher compared to room temperature [45]. All together, we therefore may estimate the triplet decay times for the majority of our copolymers in the blood stream (at saturation 20–80%) between 5 and 15 µs. The final signal will be a mixture of all these values. 

In normal tissue, the oxygenation of the blood is likely higher and for PS exhibiting the EPR effect, there will be no PDT-induced oxygen consumption in the tissue outside the vessel. Consequently, PS phosphorescence from normal tissue is much lower compared to tumor tissue, even at high pulse rates. In [17], we proposed to use this effect for tumor diagnostics. Here, we report a feasibility test of a non-contact 2D scan. A condenser lens in front of the combined optical fiber bundle projects the excitation light from the central fiber at the mouse skin. The spot had around 1 mm in diameter. Due to strong scattering, the illuminated area is much bigger than the focus. This allows usage of the same lens to funnel the emitted light from these areas to the detection fibers. This arrangement results in lower illumination intensity at the observed areas but makes the detection less sensitive to distance variations that occur when using just a 2D scan at fixed height. 

The scanned area is shown in Figure 7 on the left. It includes healthy tumor tissue, one leg, and even some pixels outside the mouse. In the middle, the total count for every pixel in the time-range 40–80 µs is encoded in grey scale with a lower cutoff at 10% of the maximum detected value.

The tumor (blue encircled in the left image) can clearly be distinguished from healthy tissue. The green encircled spot is no tumor but a vessel damage caused by intensive illumination for a longer measurement the day before. It is important to understand that this intended diagnostic detects PS phosphorescence from extravasated dye molecules, but it does not distinguish between EPR leakage and other damages. Typical signals from areas with and without extravasated dye (color coded) are shown on the right. Already this simple feasibility test yields high contrast. However, the optimization of the drug circulation time, illumination intensity, and the implementation of 3D scanning offer many options for improvement, potentially making this detection method a candidate for clinical application. 

## 4. Conclusions

We were able to give experimental evidence for the existence of a threshold level for PDT illumination intensity below which ^1^O_2_ is generated in all the tumor tissue. If the intensity is higher than this threshold and thus the PDT-induced oxygen consumption exceeds the supply, ^1^O_2_ generation is limited to blood vessels and their walls. The advantages and disadvantages of either scenario are the subject of future investigations. The detection technology presented here is a prerequisite to do that. In this context, we detected phosphorescence that could be assigned to ^1^O_2_ generated in the tumor tissue outside the vessel walls for the first time when we kept the illumination below the threshold. Furthermore, we could demonstrate that the phosphorescence of extravasated PS caused by anoxia at illumination above the threshold is a promising diagnostic tool for tumor detection, if the PS delivery system exploits the EPR effect.

## Figures and Tables

**Figure 1 jpm-13-00781-f001:**
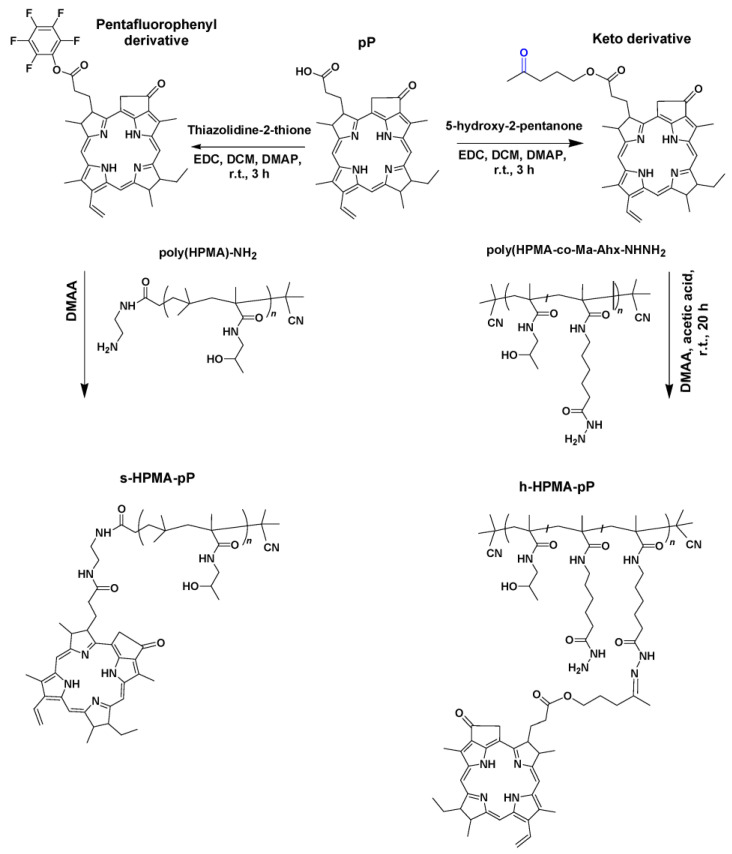
The synthesis of copolymers we used in this work. The non-cleavable s-HPMA-pP carries one pP moiety each (**left**), while the cleavable h-HPMA-pP carries 3–4 moieties each (**right**).

**Figure 2 jpm-13-00781-f002:**
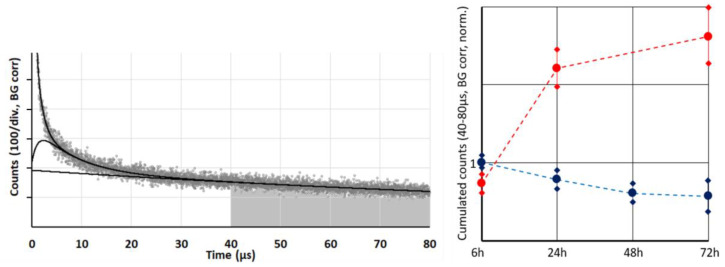
Typical *in vivo* phosphorescence signal at 1270 from mouse sarcoma after systemic application of our copolymers (**left**). The grey area, hence the cumulated counts in the time-window 40–80 µs serves as measure for the concentration of active PS in the tissue, if the illumination intensity is kept constant. The time trend of this parameter including error margins is shown on the **right** (s-HMPA-pP—red/h-HPMA-pP—blue). All values are normalized to h-HPMA-pP after 6 h.

**Figure 3 jpm-13-00781-f003:**
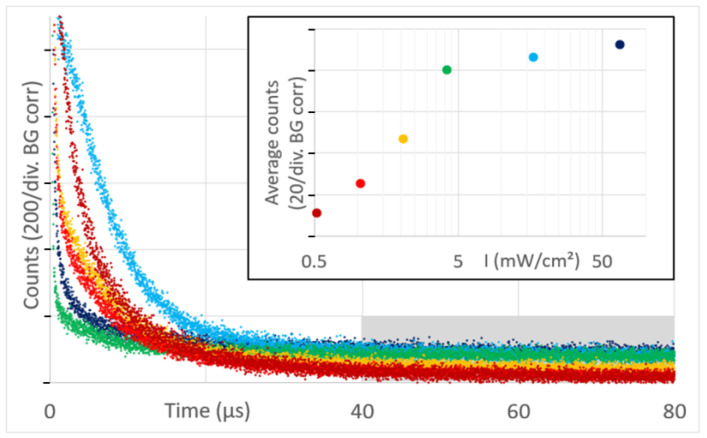
Time-resolved phosphorescence detected from mouse sarcoma, color-encoded for the illumination intensities from 0.5 to 60 mW/cm² (background corrected), hence at pulse rates between 94 Hz and 12 kHz. The insert shows the average counts in the grey marked time window for the different illumination intensities.

**Figure 4 jpm-13-00781-f004:**
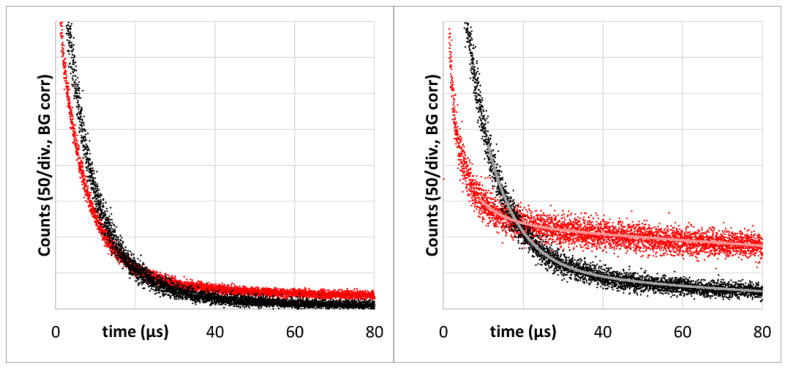
Time-resolved luminescence at 1270 nm—mouse sarcoma 72 h after systemic application of 5 mg/kg, h-HPMA-pP (**left**) or s-HPMA-pP (**right**), 360,000 Pulses at 12 kHz (red) and 375 Hz (black).

**Figure 5 jpm-13-00781-f005:**
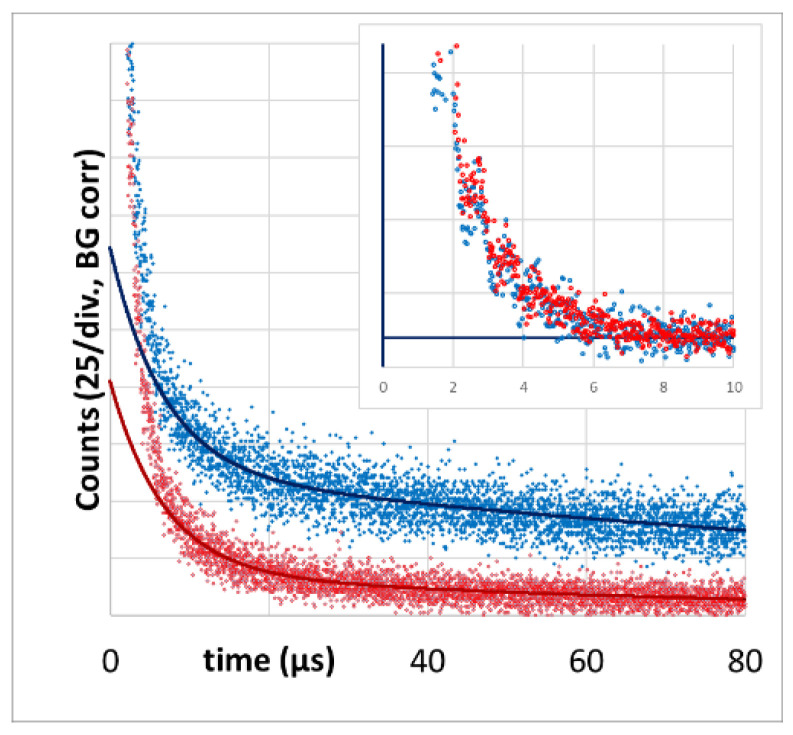
Time-resolved luminescence in the wavelength area 850–950 nm as detected from mouse sarcoma 72 h after systemic application of 5mg/kg, s-HPMA-pP, 360.000 Pulses at 12 kHz (blue) and 375 Hz (red). The insert shows the remaining identical STA after subtraction of the according two exponential tail fits.

**Figure 6 jpm-13-00781-f006:**
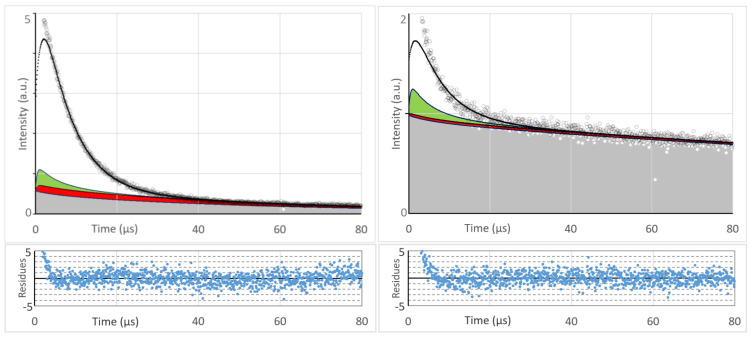
Comparing the experimental time-resolved luminescence at 1270 nm from mouse sarcoma (grey circles) with kinetics calculated based on our Krough model for the tumor (black) [17]. In case of the theoretical description, signal contributions of ^1^O_2_ in the blood (white), vessel wall (green), tumor tissue (red), and PS phosphorescence (grey) can be distinguished. 72 h after systemic application of 5 mg/kg, s-HPMA-pP. 360,000 Pulses at 375 Hz (**left**) and 12 kHz (**right**). The intensity units are identical in both diagrams; hence, the scale is stretched by a factor of 2.5 in the right diagram.

**Figure 7 jpm-13-00781-f007:**
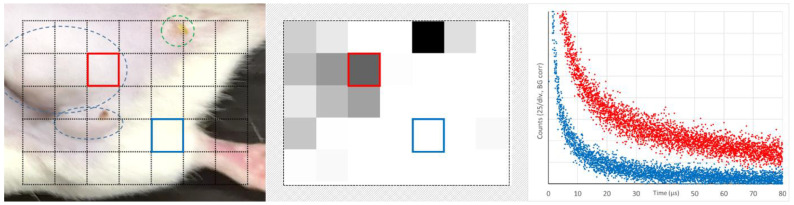
Feasibility test of tumor diagnostic using the illumination intensity above the threshold and cumulating the detection counts between 40 and 80 µs in the range 850–950 nm. Scanned area (**left**), grey-coded cumulated counts (**middle**), and detected kinetics at the two selected positions as indicated by color (**right**).

## Data Availability

Data available on request.

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
