# Peer review of "Singlet Oxygen In Vivo: It Is All about Intensity—Part 2"

_jpm, 2023, doi:10.3390/jpm13050781_

Round 1

Reviewer 1 Report

The work "Singlet Oxygen in vivo: It is all about intensity - part 2" is an interesting work following the first part of experiment for high intensity. This work could be potential for publication after some minor revisions.

1. The method section and results section are mixing-up. Please revise accordingly.

2. The result section should be describe in detail and follow by the discusion.  

Author Response

Thanks for your time and the work done to help us improve the manuscript.

For a more structured reply we included a pdf.

best wishes,

Steffen Hackbarth

Reviewer 2 Report

The authors’ report “Singlet Oxygen in vivo: It is all about intensity - part 2” comprises meticulous investigation on induced anoxia as a limiting factor for photodynamic tumor therapy (PDT). They demonstrate by modeling and experimental approaches that with illumination intensities above a certain threshold, singlet oxygen – a chemical specimen pertinent to the PDT efficiency – production is limited to the blood vessel and nearest vicinity; lower intensities allow singlet oxygen generation also in tissue which is a few cell layers away from the vessels. The authors report experimental results for intensities at both sides of the threshold and constitute proof for the described model. Ultimately the described analysis allows for better optimization and coordination of PDT drugs and treatment, as well as new diagnostic methods based on gated PS phosphorescence, as scrutinized at the end of the manuscript.

The study “Singlet Oxygen in vivo: It is all about intensity - part 2” is well carried out, and holds merit to engage the JPM readership and potentially larger audiences as well. Additionally, the manuscript is well placed in the context by citing relevant scholarly literature throughout. I suggest a few edits in order to improve the overall scientific approach of the work.

Section “1. Introduction”

- Please add 2-3 literature references to back up the statement “Today, there is a general agreement that local generation …”

- Why have the authors chosen to specifically point out Hamamatsu’s NIR-PMT?

Section “3. Results and Discussion”

- The sentence “However, we may assume that the average phosphorescence quantum yield of tissue located PS under anoxia is independent from the selected cell.” needs further clarification and/or citation to scholarly literature

- Related to Fig. 3, the authors state that “ As the overall illumination dose is the same in every measurement, this is a direct indicator for reduced PDT-induced anoxia at low illumination intensity.” While this holds true by visual inspection of the signals in Fig. 3, how significant are the differences between different illumination intensities? In the time window 40 – 80 µs these appear eventually very similar.

- The results related to Fig. 4, the authors note “This should not be seen too serious”. Please re-word this statement to a more scientific expression.

-  With respect to the decay time analyses, the authors state “Assuming, 240 μs represents the phosphorescence lifetime of pP in tissue in the absence of oxygen and taking into account the phosphorescence decay time determined in cells in vitro at 37°C, stirred under 10% oxygen (3.5 +/-0.2 μs) [xxxviii], we may estimate the local pO2 in the tissue at 375 Hz pulse rate corresponding to 78 μs decay time to be around 0.003 atm or 300 Pa.” Please elaborate, how do you calculate this 0.003 atm (or 300 Pa) correspondence?

- In the analysis with respect to Fig. 6. (especially the paragraph starting “Fig 6 left shows the comparison of theory and experiment for …”), do the authors have e.g. a pharmacokinetic (compartment) model to demonstrate the PS distribution is even in the tissue and wall? Without any PK data, how do they know, which portion of the drug is bound to blood proteins, i.e. the free vs. protein bound fraction of the drug? These factors may have impact on the results. Please elucidate with a relevant pharmacokinetic model and citations to scholarly literature and/or small sample-sized PK study.

English language is adequate throughout the manuscript.

Author Response

(The authors gave the same response as above.)
